# Emergent Hydrodynamics in an Exclusion Process with Long-Range Interactions

**A. Zahra**[1,2†]**, J. Dubail**[1,3]**, and G. M. Schütz**[2⋆]

**1** Laboratoire de Physique et Chimie Théoriques, University of Lorraine, Nancy, France
**2** Centro de Análise Matemática, Geometria e Sistemas Dinâmicos, Departamento de Matemática, Instituto Superior Técnico, Universidade de Lisboa, Lisbon, Portugal
**3** Centre Européen de Sciences Quantiques and ISIS (UMR 7006), University of Strasbourg and CNRS, Strasbourg, France

† ali.zahra@univ-lorraine.fr,   ⋆ gunter.schuetz@tecnico.ulisboa.pt

## Abstract

We study the *symmetric Dyson exclusion process* (SDEP)—a lattice gas with exclusion and long-range, Coulomb–type interactions that emerge both as the maximal-activity limit of the symmetric exclusion process and as a discrete version of Dyson's Brownian motion on the unitary group. Exploiting an exact ground-state (Doob) transform, we map the stochastic generator of the SDEP onto the spin-$\frac{1}{2}$ XX quantum chain, which in turn admits a free-fermion representation. At macroscopic scales we conjecture that the SDEP displays *ballistic* (Eulerian) scaling non-local hydrodynamics governed by the equation

$$\partial_t \rho + \partial_x j[\rho] = 0, \qquad j[\rho](x,t) = \frac{1}{\pi} \sin(\pi \rho(x,t)) \, \sinh(\pi \mathcal{H} \rho(x,t)),$$

where $\mathcal{H}$ is the Hilbert transform, making the current a genuinely non-local functional of the density. This non-local one-field description is equivalent to a local two-field "complex Hopf" system for finite particle density. Closed evolution formulas allow us to solve the melting of single- and double-block initial states, producing limit shapes and arctic curves that agree with large-scale Monte-Carlo simulations. The model thus offers a tractable example of emergent non-local hydrodynamics driven by long-range interactions.

## 1  Introduction

Stochastic interacting particle systems [1–3] are classical Markov processes defined on a lattice where particles jump with rates that depend on the position of other particles. Ergodic particle systems with one conserved species of particles on a ring with $L$ sites have for any fixed number $N$ of particles a unique stationary distribution. Because of the conservation of particles, the expected number $\rho_k(t)$ of particles on a site $k$ at time $t$ satisfies for any such process the discrete continuity equation

$$\frac{\mathrm{d}}{\mathrm{d}t} \rho_k(t) = j_{k-1}(t) - j_k(t) \tag{1}$$

where the expectation is taken w.r.t. the distribution of the particle configuration at microscopic time $t$, starting at $t = 0$ from some given initial distribution. The current

$j_k(t)$ is the average net number of particles that cross the lattice bond $(k, k+1)$ in an infinitesimal time interval $dt$. It is given by the expectation of the so-called instantaneous current $j_k^{inst}$ which is a random number that is given by the action of the infinitesimal generator of the Markov process on the (random) particle number $n_k(t)$ at time $t$.

The discrete continuity equation (1), even though mathematically exact, does not in general provide much insight into the physics of the system such as e.g. the large-scale behaviour of the evolving particle density as a function of space and time, phase transitions, and other phenomena that characterize the macroscopic state of the system in the thermodynamic $L \to \infty$. The reason for this failure is twofold.

First of all, the discrete continuity equation cannot, in general, be solved exactly. This can be seen by noting that generically the expectation of the instantaneous current involves correlations of higher order. Taking further time-derivatives of these correlations, one is usually faced with the infinite BBGKY hierarchy of equations [1]. Thus usually it is not possible to obtain a closed system of equations for a finite set of observables.

The second reason arises from the fact that in general the expectation of a local quantity, which is an average over initial states and over infinitely many realizations of the stochastic dynamics, does not automatically describe the typical behaviour of the particle system observed on coarse-grained scale in a *single* realization of the stochastic dynamics. This latter problem, however, is what interests us: When observing, e.g., a piston filled with a gas, we do *not* wish to compute how the gas expands under heating on average in a very large number of experiments. We want to compute how it behaves in any single experiment, corresponding to a single realization of the underlying effectively stochastic dynamics. This is possible because on macroscopic scale the gas behaves deterministically always in the same way. Hence averaging over realizations of the process does not provide new information, studying the large-scale behaviour of a large number of atoms in a single realization is sufficient.

For a variety of stochastic interacting particle systems with one conserved species of particles these two problems have been overcome mathematically rigorously by establishing that the large scale evolution of the *coarse-grained* particle density $\rho(x, t)$ is governed by the macroscopic continuity equation

$$\partial_t \rho(x, t) + \partial_x j(x, t) = 0 \tag{2}$$

where in spatially homogeneous systems the current is typically of the form

$$j(x, t) = j^*(\rho(x, t)) - D(\rho(x, t))\partial_x \rho(x, t). \tag{3}$$

Here $j^*(\cdot)$ is the stationary current density relation which like the collective diffusion coefficient $D(\cdot)$ depend only on the local density $\rho(x, t)$ [1, 2]. Thus the hydrodynamic equation (2) is a closed (in general nonlinear) deterministic partial differential equation for the macroscopic density profile $\rho(x, t)$. The macroscopic space point $x$ represents a rescaled microscopically large region of the underlying lattice on which the microscopic model is defined. More precisely, this rescaling amounts to introducing a lattice constant $a$ and looking at the particle system in a region on the lattice of length $na$ around some site. In the limit $a \to 0$ this lattice site and the region around it then represent the space point $x$. The macroscopic time is also large compared to the microscopic time and usually taken with a rescaling $t \to t/a^x$ for some suitably chosen scaling exponent $x$ that depends on the particle system under consideration.

The emergence of the macroscopic continuity equation (2) from the microscopic discrete continuity equation (1) rests on two generic properties of stochastic interacting particle systems. The first is the law of large numbers which asserts that the random number of particles in a large region (i.e., a large segment as described above) converges to a

deterministic density $\rho(x,t)$ times the size of the segment. The second reason is local stationarity which means that the system is locally in the stationary state at density $\rho(x,t)$ which results from rescaling the microscopic time $t$ to become very large. Since in one dimension stationary correlations are generically short ranged [4], the law of large numbers can indeed by applied to describe the system locally (on large scale) by a density $\rho(x,t)$ and the current $j(x,t)$ depends on $x$ and $t$ only through stationary expectations which can be expressed as functions of $\rho(x,t)$.

Local stationarity also implies that the microscopic details of the initial distribution are washed out and do not matter. Therefore one expects that only the coarse-grained initial density $\rho_0(x) := \rho(x,0)$ determines the coarse-grained density $\rho(x,t)$ at a later macroscopic time $t$. Since the two properties of the law of large numbers in a large region and local stationarity are generic, the hydrodynamic limit (2) is expected to be valid generically, even when there is no rigorous mathematical proof [1]. Indeed, even when – like in boundary-driven systems – correlations are not short-ranged but sufficiently weak, the hydrodynamic description is still valid and can proved rigorously for some classes of stochastic interacting particle systems [5–7].

This picture is well-established for particle systems with short-range interactions. It cannot, however, be applied naively to stochastic interacting particle systems with long-interactions, for which exact results are scarce [8–11] and where the notion of locality of the current, i.e., its dependence as a function of the local density, becomes open to debate. To address this issue and show that this locality may indeed get lost, we compute in this work the large scale dynamics of a particular stochastic interacting particle system with long-range interactions, viz., a symmetric exclusion process where the transition rates depend via a logarithmic pair potential not only on the occupation of nearest neighbor sites, but on the entire configuration of the system.

As elaborated below, this process arises in similar form in a wide variety of contexts and it may thus be seen as paradigmatic. Using tools from condensed matter theory we show that at large space- and time scales the expected local density appearing in the discrete continuity equation (1) is governed by a closed hydrodynamic equation of the form (2) where, however, the current $j(x,t)$ is *not* a local function of the density $\rho(x,t)$, but a functional that via a Hilbert transform depends on the complete density profile at all points $y$ in space. This demonstrates that local stationarity remains crucial also in the presence of long-range interactions, but with the generalization that local variables are determined by the stationary various everywhere instead of only locally.

This paper is organized as follows. In Sec. 2 we introduce the Symmetric Dyson Exclusion Process and point out some of its origins and connections to related problems. The stochastic dynamics is defined in terms of the intensity matrix [3,12,13] whose off-diagonal elements are the transition rates and whose diagonal elements guarantee conservation of probability. In Sec. 3 we formulate the emergent macroscopic description: starting from the exact microscopic expressions we propose and motivate a *non-local hydrodynamic equation*. We discuss its equivalence to a local two-field (complex Hopf) hydrodynamic equation and its relation to other works. Section 4 tests these predictions against large-scale Monte-Carlo simulations. We treat the "melting" of single- and double-block initial states. We uncover the emergence of a space-time limit shape phenomenon that are characteristic to the model, Fig. 2, and derive explicit arctic curves, and find excellent agreement with numerical data.

# 2 Symmetric Dyson Exclusion Process (SDEP)

We consider the symmetric Dyson exclusion process, denoted below by the acronym SDEP, which is an exclusion process with nearest neighbor jumps and a long-range interaction that arises in various seemingly unrelated contexts. This process was first introduced as a model for steps of a vicinal surface in [14] and later independently in [15, 16] by maximally conditioning the conventional Symmetric Simple Exclusion Process (SSEP) on a large number of particle jumps following the approach of [17, 18]. For the convenience of the reader, we provide a brief overview of this conditioning in the Appendix. The invariant measure of the SDEP, reviewed below, is described by the distribution of eigenvalues of random unitary matrices that perform Dyson's Brownian motion over the unitary group $U(N)$, thus making the SDEP a discrete interacting random walk version of Dyson's Brownian motion [19]. It is also closely related to classical point charges on a circle with a repulsive force arising from the two-dimensional Coulomb law. Endowed with Brownian motion, the dynamics of these point charges is intimately related to the Calogero–Sutherland model [20, 21] which has connections to the Kadomtsev–Petviashvili equation [22, 23].

Importantly, by conditioning the SSEP on an atypical activity, i.e. on an atypical number of particle jumps, the intensity matrix of the conditioned process is, up to an energy shift, the ground state transformation of the ferromagnetic Heisenberg spin-1/2 XXZ quantum chain which exhibits a phase transition from a phase-separated domain wall state [24] to a hyperuniform conformally invariant phase [25, 26]. The phase transition occurs at a disorder point that corresponds to the unconditioned process, i.e., the conventional SSEP [12, 27, 28]. When conditioned to maximal activity, the intensity matrix is the ground state transform of the Hamiltonian of the spin-1/2 XX quantum chain (see Eq. (18)), denoted below by $H^{\mathrm{XX}}$.

## 2.1 Definition of the SDEP

In the conventional SSEP defined on a chain of $L$ sites with periodic boundary conditions, each lattice site $k$ of a lattice of $L$ sites is occupied by at most particle. Particles in an ergodic sector with $N$ particles are labelled sequentially by integers $i \in \{1, \ldots, N\}$. The $i$-th particle at position $k_i$ attempts to jump after an exponential random time with parameter $w$ to a neighboring site $k \pm 1$ with equal probability 1/2. If the selected target is empty then it jumps, otherwise the jump attempt is rejected and the particle stays at site $k_i$, thus respecting the on-site exclusion interaction. Below we label a configuration with particles at sites $k_1, \ldots, k_N$ by $\boldsymbol{k}$.

In the SDEP the interaction is long-ranged. The $i$-th particle at position $k_i$ attempts to jump to the neighboring site $k_i \pm 1$ with the rate

$$w_N^{\pm}(i) = \frac{w}{2} \prod_{\substack{j=1 \\ j \neq i}}^{N} \frac{\sin \pi \frac{k_j - k_i \mp 1}{L}}{\sin \pi \frac{k_j - k_i}{L}} \tag{4}$$

which depends on the position of all other particles on the lattice. The factor $w$ sets the time scale of the process. The corresponding instantaneous current, i.e., the net number of particle jumps across the lattice bond $(k, k+1)$ in an infinitesimal time interval, is given by

$$j_k^{\mathrm{inst}} = \frac{w}{2} \sum_{i=1}^{N} \left( \delta_{k_i, k} \prod_{\substack{j=1 \\ j \neq i}}^{N} \frac{\sin \pi \frac{k_j - k_i - 1}{L}}{\sin \pi \frac{k_j - k_i}{L}} - \delta_{k_i, k+1} \prod_{\substack{j=1 \\ j \neq i}}^{N} \frac{\sin \pi \frac{k_j - k_i + 1}{L}}{\sin \pi \frac{k_j - k_i}{L}} \right). \tag{5}$$

The process is reversible w.r.t. the invariant measure [14, 15]

$$\pi_N^*(\boldsymbol{k}) = \frac{1}{Z_N} e^{-V_N(\boldsymbol{k})} \tag{6}$$

with the long-range interaction potential

$$V_N(\boldsymbol{k}) = -\sum_{i=1}^{N-1} \sum_{j=i+1}^{N} \ln\left(\sin^2\left(\pi \frac{k_j - k_i}{L}\right)\right) \tag{7}$$

and where $Z_N$ is a normalization factor [15]. The above-mentioned link with the eigenvalues $\exp(2\pi i x_j)$ of random unitary matrices that perform Dyson's Brownian motion over the unitary group $U(N)$ becomes apparent by the identification $k_j := x_j L$ of the parameters $x_j$ with the rescaled particle coordinates $k_j$. However we stress that here the eigenvalues are restricted to the discrete set of $L^{\text{th}}$ roots of unity.

## 2.2 Some useful facts about the spin-1/2 XX quantum chain

We label the lattice sites by integers $k \in \{-L/2 + 1, \ldots, L/2\}$ modulo $L$. This is to facilitate taking the two-sided thermodynamic limit $L \to \infty$. The spin-1/2 XX quantum chain is defined by the Hamiltonian [29]

$$H^{\text{XX}} = -\frac{w}{4} \sum_{k=-L/2+1}^{L/2} \left(\sigma_k^x \sigma_{k+1}^x + \sigma_k^y \sigma_{k+1}^y\right) \tag{8}$$

and can be turned by a Jordan-Wigner transformation $\sigma_k^+ = \prod_{j=-L/1+1}^{k-1} (-1)^{c_j^\dagger c_j} c_k^\dagger$ into a system of spinless free fermions with fermionic creation and annihilation operators $c_k^\dagger$, $c_k$ that satisfy the anticommutation relations $\{c_k^\dagger, c_l\} = \delta_{k,l}$,

$$H^{\text{XX}} = -\frac{w}{2} \sum_{k=-L/2+1}^{L/2} \left(c_k^\dagger c_{k+1} + c_{k+1}^\dagger c_k\right). \tag{9}$$

For self-containedness we recall some well-known properties of the XX-chain, see e.g. [29, 30] for details. The $N$-particle states are defined by the ordered product

$$|\boldsymbol{k}\rangle := \prod_{n=1}^{\overset{N}{\rightarrow}} c_{k_n}^\dagger |\emptyset\rangle = c_{k_1}^\dagger c_{k_2}^\dagger \ldots c_{k_N}^\dagger |\emptyset\rangle \tag{10}$$

where $|\emptyset\rangle$ is the vector representing the empty lattice (or all spins up in spin language). Ordering means that for site indices $i \in \{1, \ldots, N\}$ the particle locations $k_i$ in $\boldsymbol{k}$ are from the subset $\Omega_N$ of $\{-L/2 + 1, \ldots, L/2\}^N$ defined by $k_j > k_i$ for $j > i$. The projection operator

$$\hat{P}_N := \sum_{\boldsymbol{k} \in \Omega_N} |\boldsymbol{k}\rangle\langle\boldsymbol{k}| \tag{11}$$

yields the Hamiltonian

$$H_N^{\text{XX}} = H^{\text{XX}} \hat{P}_N \tag{12}$$

of the $N$-particle sector. The (normalized) ground state vector $|0_N\rangle$ is well known [30],

$$|0_N\rangle = \sum_{\boldsymbol{k}} \Psi_N(\boldsymbol{k}) |\boldsymbol{k}\rangle \tag{13}$$

with the ground state wavefunction

$$\Psi_N(\boldsymbol{k}) = \frac{2^{\binom{N}{2}}}{L^{N/2}} \prod_{i=1}^{j} \prod_{j=i+1}^{N} \sin\left(\pi \frac{k_j - k_i}{L}\right). \tag{14}$$

The corresponding ground state energy is [30]

$$E_N = -w \frac{\sin\left(\frac{\pi N}{L}\right)}{\sin\left(\frac{\pi}{L}\right)} \tag{15}$$

and the spectral gap corresponding to an excitation of momentum $2\pi/L$ is

$$\Delta E_N = 2w \sin\left(\frac{\pi N}{L}\right) \sin\left(\frac{\pi}{L}\right) \simeq \frac{2\pi}{L} w \sin\left(\frac{\pi N}{L}\right), \tag{16}$$

from which one reads off the sound velocity

$$v_s = w \sin\left(\pi \rho\right) \tag{17}$$

for density $\rho = N/L$.

## 2.3 Intensity matrix of the SDEP

For $N$ particles the intensity matrix $H_N$ of the SDEP is the Doob transform of the Hamiltonian of the XX-chain,

$$H_N := \hat{\Psi}_N \left(H_N^{\text{XX}} - E_N\right) \hat{\Psi}_N^{-1}, \tag{18}$$

where $\hat{\Psi}_N$ is the diagonal matrix

$$\hat{\Psi}_N := \sum_{\boldsymbol{k} \in \Omega_N} \Psi_N(\boldsymbol{k}) |\boldsymbol{k}\rangle\langle\boldsymbol{k}|. \tag{19}$$

Probability conservation is encoded in the $N$-particle summation vector

$$\langle s_N | = \sum_{\boldsymbol{k} \in \Omega_N} \langle \boldsymbol{k}| \tag{20}$$

by the eigenvalue property $\langle s_N | H_N = 0$. The invariant measure is represented by the vector

$$|\pi_N\rangle := \sum_{\boldsymbol{k} \in \Omega_N} \Psi_N^2(\boldsymbol{k}) |\boldsymbol{k}\rangle = \hat{\Psi}_N^2 |s_N\rangle. \tag{21}$$

Note that due to reversibility of the process the average value of the current in the steady state is

$$\langle \sigma_k^+ \sigma_{k+1}^- - \sigma_k^- \sigma_{k+1}^+ \rangle = 0 \tag{22}$$

while for the average value of the activity one finds from [30]

$$\frac{w}{2} \langle \sigma_k^+ \sigma_{k+1}^- + \sigma_k^- \sigma_{k+1}^+ \rangle = \frac{w}{L} \frac{\sin(\frac{\pi N}{L})}{\sin(\frac{\pi}{L})} \xrightarrow[L \to \infty]{} \frac{w}{\pi} \sin(\pi \rho) \tag{23}$$

where the thermodynamic limit is taken with a fixed density $\rho = \frac{N}{L}$. This result for the average activity at equilibrium in the SDEP will play an important role below.

# 3    Hydrodynamics

We now turn to our main goal, which is to derive the coarse-grained hydrodynamics of the SDEP. In this Section we make a precise conjecture regarding the form of the hydrodynamic limit of the SDEP, which will be supported by numerical evidence presented in Sec. 4.

Note that, in the conventional SSEP, the spectral gap of the intensity matrix scales as $O(1/L^2)$ corresponding to diffusive scaling with dynamical exponent $z = 2$. This is of course consistent with the fact that the hydrodynamic limit of the SSEP is given by the heat equation [3]. In contrast, the spectral gap of the intensity matrix of the SDEP scales as $O(1/L)$, see Eq. (16), which leads to faster relaxation with dynamical exponent $z = 1$. We thus expect that, upon coarse-graining under ballistic (Eulerian) scaling the SDEP satisfies a hydrodynamic equation resulting from the continuity equation

$$\partial_t \rho(x, t) + \partial_x j(x, t) = 0 \tag{24}$$

with a current $j$ that turns out to be a functional of $\rho(x, t)$. Indeed, our main claim is that the exact form of the current is

$$j[\rho](x, t) = \frac{w}{\pi} \sin(\pi \rho(x, t)) \sinh(\pi \mathcal{H} \rho(x, t)), \tag{25}$$

where $\mathcal{H} f(x, t)$ is the Hilbert transform w.r.t. the first argument of a function $f(x, t)$ that is periodic in $x$ with period $L$, defined as the principal value integral

$$\mathcal{H} f(x, t) = \text{p.v.} \frac{1}{L} \int_0^L \frac{f(u, t) du}{\tan \frac{\pi(x-u)}{L}}. \tag{26}$$

Since the Hilbert transform is not local, we see that the particle current (25) is not local.

We will provide compelling numerical evidence supporting this claim in Sec. 4. For now, let us motivate the form (25) of the current, and discuss the relation between this claim and the existing literature.

## 3.1    Motivation for the conjectured particle current (25)

Let us start from the exact expression for the instantaneous current in the SDEP, Eq. (5). For a given configuration of particles at positions $k_1, \ldots, k_N$, we define the instantaneous density of particles on site $k$ as $n_k^{\text{inst}} = \sum_{j=1}^N \delta_{k, k_j} \in \{0, 1\}$. We rewrite formula (5) as

$$
\begin{aligned}
j_k^{\text{inst}} &= \frac{w}{2} n_k^{\text{inst}} (1 - n_{k+1}^{\text{inst}}) \exp \left[ \sum_{q \neq k, k+1} n_q^{\text{inst}} \log \left( \frac{\sin \pi \frac{q-k-1}{L}}{\sin \pi \frac{q-k}{L}} \right) \right] \\
&\quad - \frac{w}{2} (1 - n_k^{\text{inst}}) n_{k+1}^{\text{inst}} \exp \left[ - \sum_{q \neq k, k+1} n_q^{\text{inst}} \log \left( \frac{\sin \pi \frac{q-k-1}{L}}{\sin \pi \frac{q-k}{L}} \right) \right].
\end{aligned} \tag{27}
$$

This expression is exact. Now we start making approximations. First, we can approximate the sum in the two exponentials in terms of the coarse-grained density $\rho(x)$,

$$
\sum_{\substack{q=0 \\ q \neq k, k+1}}^{L-1} n_q^{\text{inst}} \log \left( \frac{\sin \pi \frac{q-k-1}{L}}{\sin \pi \frac{q-k}{L}} \right) \underset{L \gg 1}{\simeq} \left( \int_0^{k-\epsilon} + \int_{k+\epsilon}^L \right) \log \left( \frac{\sin \pi \frac{x-k-1}{L}}{\sin \pi \frac{x-k}{L}} \right) \rho(x) dx
$$

$$
\simeq \quad \text{p.v.} \int_0^L \log \left( 1 - \frac{\pi}{L} \cotan \pi \frac{x-k}{L} \right) \rho(x) dx
$$

$$\simeq \quad \pi \, \mathcal{H}\rho(k) \, .$$

When we make this approximation, we expect that we are correctly describing the contribution of the long-distance terms with $|q - k| \gg 1$. However we are overlooking details at short distances, where $q$ is distant from $k$ from only a few lattice sites.

This suggests a second approximation: that the ratio

$$\frac{w}{2} \, \frac{n_k^{\text{inst}}(1 - n_{k+1}^{\text{inst}}) \exp\left[\sum_{q \neq k, k+1} n_q^{\text{inst}} \log\left(\frac{\sin \pi \frac{q-k-1}{L}}{\sin \pi \frac{q-k}{L}}\right)\right]}{\exp(\pi \mathcal{H}\rho(k))}$$

depends only on the local state of the gas. This is a mean-field approximation. Upon coarse-graining, that ratio should become a function of the local particle density $\rho(x, t)$ only. Thus, a natural Ansatz for the coarse-grained particle current in the SDEP is

$$j[\rho](x, t) = a_+(\rho(x, t))e^{\pi \mathcal{H}\rho(x, t)} - a_-(\rho(x, t))e^{-\pi \mathcal{H}\rho(x, t)}, \tag{28}$$

for some functions $a_+(\cdot)$ and $a_-(\cdot)$.

Because of reflexion symmetry, we must have $a_+(\cdot) = a_-(\cdot)$. Moreover, if we look back at formula (27) and replace the minus sign between the two terms by a plus sign, we get the instantaneous local activity $a_k^{\text{inst}}$ instead of the particle current $j_k^{\text{inst}}$. So the coarse-grained activity should be

$$a[\rho](x, t) = a_+(\rho(x, t))e^{\pi \mathcal{H}\rho(x, t)} + a_-(\rho(x, t))e^{-\pi \mathcal{H}\rho(x, t)}. \tag{29}$$

Now we appeal to local equilibrium. When we apply this formula to an equilibrium state, which has a constant density $\rho(x, t) = \rho$, we have $\mathcal{H}\rho(x, t) = 0$, and so we find that $a_+(\rho) + a_-(\rho)$ is the activity at equilibrium. Thus, we conclude that the coarse-grained activity and the coarse-grained current should be given respectively by

$$\begin{aligned}
a[\rho](x, t) &= a_{\text{eq}}(\rho(x, t)) \cosh(\pi \mathcal{H}\rho(x, t)), \\
j[\rho](x, t) &= a_{\text{eq}}(\rho(x, t)) \sinh(\pi \mathcal{H}\rho(x, t)),
\end{aligned} \tag{30}$$

where $a_{\text{eq}}(\rho)$ is the local activity at equilibrium. The local activity at equilibrium should be identified with the exact lattice result (23), which leads to the conjectured form of the current (25).

## 3.2 Equivalence between non-local one-component hydrodynamics and local two-component hydrodynamics

From now on we set the time scale $w = 1$ to lighten the formulas.

It turns out that, because of the mathematical properties of the Hilbert transform, it is possible to reformulate the *non-local* one-component hydrodynamic equation

$$\partial_t \rho(x, t) + \partial_x\left[\frac{1}{\pi} \sin(\pi \rho(x, t)) \sinh(\pi \mathcal{H}\rho(x, t))\right] = 0 \tag{31}$$

as a *local* two-component hydrodynamic equation, with two densities $\rho(x)$ and $\tilde{\rho}(x)$ that can be viewed as independent,

$$\begin{cases}
\partial_t(\pi \rho(x, t)) + \partial_x[\frac{1}{\pi} \sin(\pi \rho(x, t)) \sinh(\pi \tilde{\rho}(x, t))] &= 0 \\
\partial_t(\pi \tilde{\rho}(x, t)) + \partial_x[\frac{1}{\pi} \cos(\pi \rho(x, t)) \cosh(\pi \tilde{\rho}(x, t))] &= 0 \, .
\end{cases} \tag{32}$$

Eqs. (31) and (32) are equivalent as long as the initial conditions $\rho_0(x)$ and $\tilde{\rho}_0(x)$ at time $t = 0$ satisfy

$$\tilde{\rho}_0(x) = \mathcal{H}\rho_0(x). \tag{33}$$

The reason for this equivalence is as follows (see also the appendix of Ref. [31] for a similar calculation). The fundamental property of the Hilbert transform is that $f(x)$ and $\mathcal{H}f(x)$ are the real- and imaginary parts of the same analytic function in the upper half-plane. More specifically, for a periodic function $f(x)$ of period $L$, there is exists a function $F(z)$ defined for any $\mathrm{Im}\, z > 0$, that is analytic in the upper half-plane (including at infinity), with the same periodicity as $f$, i.e. $F(z + L) = F(z)$, and such that for all $x \in \mathbb{R}$

$$F(x + i\epsilon) \underset{\epsilon \to 0^+}{=} f(x) + i\mathcal{H}f(x). \tag{34}$$

To see why this implies the equivalence between (31) and (32), we observe that, if the initial condition (33) is satisfied, then there is an analytic function $P_0(z)$ in the upper half-plane, that is periodic, and such that $P_0(x + i\varepsilon) \underset{\epsilon \to 0^+}{=} \pi\rho_0(x) + i\pi\tilde{\rho}_0(x)$. Then we consider the partial differential equation

$$\partial_t P(z, t) + i\partial_z[\cos(P(z, t))] = 0, \tag{35}$$

with the initial condition $P(z, 0) = P_0(z)$. Since the initial condition is analytic in the upper half-plane, $P(z, t)$ remains so at later times. Then we can identify the densities $\rho(x, t)$ and $\tilde{\rho}(x, t)$ with the real- and imaginary parts of $\frac{1}{\pi}P(z, t)$ along the real axis at any time,

$$\rho(x, t) = \lim_{\epsilon \to 0^+} \frac{1}{\pi}\mathrm{Re}\, P(x + i\epsilon, t), \qquad \tilde{\rho}(x, t) = \lim_{\epsilon \to 0^+} \frac{1}{\pi}\mathrm{Im}\, P(x + i\epsilon, t) \quad (= \mathcal{H}\rho(x, t)) \tag{36}$$

and we see that Eqs. (31) and (32) are both equivalent to (35) by taking the real- and imaginary parts of that equation.

## 3.3 Relation with Abanov's lattice free fermion imaginary-time hydrodynamics

The complex partial differential equation (35) is not new. In the context of large-scale descriptions of systems of hard-core particles or spin chains in one dimension, it was discussed by Abanov in Ref. [32]. In particular as pointed out by Abanov, for the quantity $\zeta(z, t) = \sin P(z, t)$ , it is equivalent to a complex Hopf (or inviscid Burgers) equation:

$$\partial_t \zeta - i\zeta\partial_z\zeta = 0. \tag{37}$$

In this form, this equation has also appeared in various other contexts, as we briefly review below in Sec. 3.4. In Ref. [32], Abanov uses this equation to study the *emptiness formation probability* of the quantum XX spin chain. This is the probability that, upon measuring simultaneously the states of all spins in an interval in the ground state of the Hamiltonian $H^{\mathrm{XX}}$, one finds that they are all pointing down. This probability is found to decay as a Gaussian as a function of the interval length, and Abanov derives several exact results about this quantity using the complex Hopf equation. For this, he relies on the fundamental fact that the ground state can be obtained through an imaginary-time evolution:

$$|\text{ground state}\rangle \propto \lim_{t \to +\infty} e^{-tH} |\psi_0\rangle \underset{t \to it}{=} \lim_{t \to -i\infty} e^{-itH} |\psi_0\rangle,$$

where $|\psi_0\rangle$ is an arbitrary state with non-vanishing overlap with the ground state.

The *real-time* hydrodynamics of the quantum XX spin chain can be easily understood, as it reflects the dynamics of the underlying non-interacting fermions [33]. One way to proceed is to think of the non-interacting fermions semi-classically. One can describe the system on large scales by its phase-space occupation $W(x, p)$, where the momentum $p$ is defined modulo $2\pi$ because the model is considered on the lattice.

Semi-classically, $W(x, p)$ is a function that takes values between 0 and $\frac{1}{2\pi}$. The local particle density is obtained by integrating over momenta at a fixed position $x$,

$$\rho(x) = \int_{-\pi}^{\pi} dp \, W(x, p) \quad \in [0, 1]. \tag{38}$$

The fermions move at their group velocity $v(p) = \sin(p)$, so $W(x, p)$ evolves in time according to the kinetic equation

$$\partial_t W + v(p)\partial_x W = 0. \tag{39}$$

The connection with the Hopf equation appears when one focuses on states that are locally equivalent to the ground state at some local density $\rho(x)$. Indeed, for such states, the phase-space occupation takes the form of a position-dependent Fermi sea, with an occupation function at position $x$ that is maximal in some interval of $p$ and vanishes in its complement:

$$W(x, p) = \begin{cases} \frac{1}{2\pi} & \text{if} \quad p \in [p_-(x), p_+(x)] \\ 0 & \text{if} \quad p \in [p_+(x), 2\pi + p_-(x)]. \end{cases} \tag{40}$$

Here $p$, $p_+$ and $p_-$ are all defined modulo $2\pi$. In such states the local density (38) is

$$\rho(x) = \frac{p_+(x) - p_-(x)}{2\pi}. \tag{41}$$

The kinetic equation (39) boils down to a Hopf-like equation for the *Fermi points* $p_\pm(x, t)$,

$$\partial_t p_\pm + \sin(p_\pm)\partial_x p_\pm = 0. \tag{42}$$

The fact that the real-time dynamics of lattice free fermions initiated in states of zero entropy is given by Eq. (42) has been studied by many authors [33–37]. Abanov's main point is that questions related to the XX chain in *imaginary time* may then be tackled by considering the Wick rotation $t \to -it$ of Eq. (42). This is precisely the equation (35) that we have encountered in the previous section, which is equivalent to the complex Hopf equation (37).

## 3.4 Relation to other works

Imaginary-time hydrodynamics of free fermions has connections with many other problems. For instance, mutually avoiding directed polymers can be mapped to the world-lines of fermions in Euclidean time, a fact pointed out by de Gennes in the 1960s [38]. The complex Hopf (or inviscid Burgers) equation (37) is also ubiquitous in problems of dimer coverings [39–41], in certain random matrix problems such as the evaluation of the asymptotics of the Harish–Chandra–Itzykson–Zuber integral [42], in studies of the emptiness formation probability [32, 43–46] and full counting statistics in spin chains [47], or in fluid two-dimensional fluid dynamics where it appears in connection with the Marangoni effect [48].

More closely related to the SDEP, a series of recent works has focused on the continuous Dyson gas of Brownian particles with Coulomb repulsion on the continuum line [49–52], with a perspective that is similar to ours. In fact the SDEP can be viewed as the *lattice*

analogue of the continuous Dyson gas; the main structural difference is the hard–core constraint $\rho \leq 1$ that comes automatically in a lattice exclusion process. For the continuous Dyson gas, the hydrodynamic equation of Refs. [49–52] reads

$$(\text{continuous Dyson gas}) \qquad \partial_t \rho + \partial_x[\pi \rho \mathcal{H} \rho] = 0. \tag{43}$$

This equation can easily be recovered from our hydrodynamic equation for the SDEP (Eq. (31)) by taking the low-density limit $\rho(x,t) \ll 1$.

## 4 Density profile and limit shape for block initial states

In this section we provide numerical checks of the validity of the hydrodynamic equation (31). We numerically simulate the microscopic dynamics of the SDEP using a kinetic Monte Carlo algorithm. For concreteness, we focus on two simple initial states: $N$ particles packed in a single block, and $N = N_1 + N_2$ particles packed into two separate blocks, see Fig. 1.

### 4.1 Density Profiles

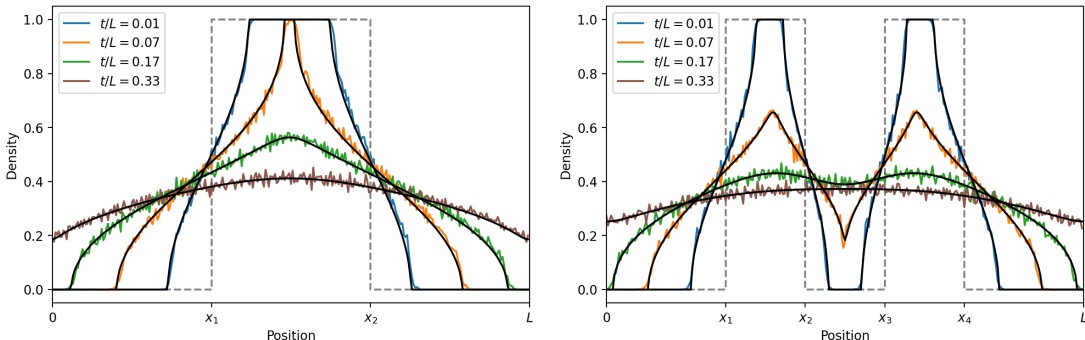

Figure 1: Evolution of the density profile from a single block initial condition (left) and from a double-block initial condition (right); the dashed gray line shows the initial density profile. We compare the result of the stochastic simulation of the SDEP for $N = 100$ particles on $L = 300$ sites with periodic boundary conditions, to the hydrodynamic prediction (black continuous lines) obtained by solving equation (45) numerically. The profiles for the stochastic evolution are averaged over 1000 independent realizations.

We start by comparing the density profiles at different times obtained from a direct stochastic simulation of the SDEP to the solution of the hydrodynamic equation (31). The Monte Carlo simulation is performed using a Gillespie-type algorithm, standard for continuous-time Markov processes. Starting from a configuration $C_{t_i}$, the next configuration $C_{t_{i+1}}$ is chosen from the set of accessible configurations with probability proportional to the transition rate from $C_{t_i}$ to $C_{t_{i+1}}$, after a waiting time $t_{i+1} - t_i$ drawn from an exponential distribution with rate equal to the sum of outgoing transition rates from $C_{t_i}$ to all accessible configurations.

To obtain the solution of the hydrodynamic equation (31), we rewrite it in the form (see Eq. (35))

$$\partial_t P(x,t) - i \sin(P(x,t)) \partial_x P(x,t) = 0 \tag{44}$$

where $P(x,t) = \pi\rho(x,t) + i\pi\mathcal{H}\rho(x,t)$. Similarly to the real inviscid Burgers equation, this equation admits solutions of the form

$$P = P_0(x + it\sin(P)), \tag{45}$$

where $P_0$ is an analytic function in the upper-half plane, determined by the initial condition at $t = 0$,

$$\text{Re}\, P_0(x + i0^+) = \pi\rho_0(x). \tag{46}$$

For the single block, corresponding to the initial condition $\rho_0(x) = 1$ if $x \in [x_1, x_2]$ and $\rho_0(x) = 0$ otherwise, we take $P_0(z)$ as

$$\text{(single block)} \qquad P_0(z) = i\,\ln\left[\frac{\sin\left(\frac{\pi}{L}(z - x_1)\right)}{\sin\left(\frac{\pi}{L}(z - x_2)\right)}\right], \tag{47}$$

while for the two blocks $[x_1, x_2] \cup [x_3, x_4]$, where $\rho_0(x) = 1$ if $x \in [x_1, x_2] \cup [x_3, x_4]$ and 0 otherwise, we have

$$\text{(two blocks)} \qquad P_0(z) = i\,\ln\left[\frac{\sin\left(\frac{\pi}{L}(z - x_1)\right)\sin\left(\frac{\pi}{L}(z - x_3)\right)}{\sin\left(\frac{\pi}{L}(z - x_2)\right)\sin\left(\frac{\pi}{L}(z - x_4)\right)}\right]. \tag{48}$$

Here 'ln' stands for the principal logarithm, with the branch cut along the negative real axis, i.e. $\ln z = \ln|z| + i\,\arg z$.

To find the solution of Eq. (45) for these two initial conditions, we use a fixed-point iteration method. We have observed that this method converges for all values of $(x, t)$ that we tested, so that one can numerically evaluate $P(x, t)$ easily. The density profile $\rho(x, t)$ is then obtained as the real part of $\frac{1}{\pi}P(x, t)$. The resulting curves for $\rho(x, t)$ are plotted in Fig. 1, and we see that they perfectly match the data points obtained from the Monte Carlo simulation of the SDEP. This confirms the validity of the conjectured hydrodynamic equation (31).

## 4.2 Limit shapes in spacetime

In Fig. 1 we see that the regions where the density is initially 0 or 1 do not immediately melt. Instead these two regions have sharp boundaries that persists up to some time. Thus, when we look at the trajectories of particles in space-time for the SDEP, one clearly sees the appearance of a *limit shape phenomenon*, see Fig. 2. We observe the emergence of a sharp boundary between 'frozen' regions where the particle density is exactly 0 or 1, and 'fluctuating' regions where the density is strictly between 0 and 1. Such limit shape phenomena occur in various statistical physics problems, including crystal growth [53,54], dimer coverings [55–57], statistics of random Young tableaux [58], the six-vertex model with domain-wall boundary conditions [40,59–61], see e.g. [62,63] for introductions. The curve separating the frozen regions and fluctuating regions is known as the arctic curve [55,64–67]. The hydrodynamic equation (31) can be exploited to make a prediction for the arctic curve for the above initial conditions, which can then be compared to the stochastic simulation of the SDEP, providing further numerical evidence for the validity of the hydrodynamic equation (31). This is what we do next.

### 4.2.1 Arctic curve for melting of a single block.

To simplify the formulas and the determination of the arctic curve, from now on we take the thermodynamic limit $L \to \infty$. In this limit and for a finite number of particles $N$, the

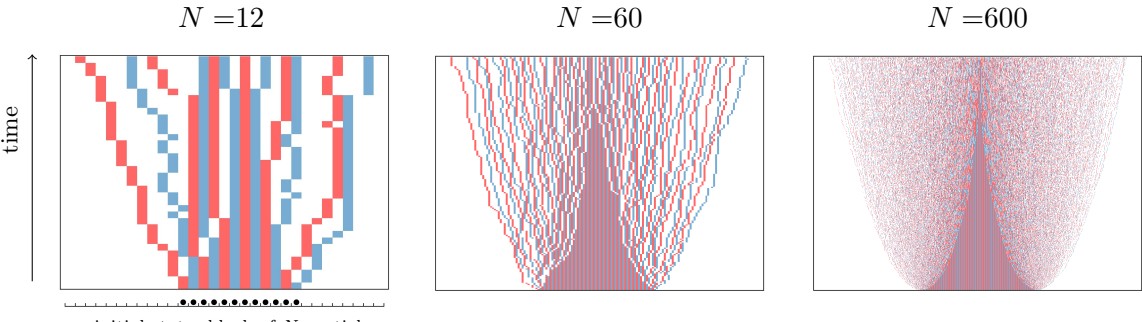

$N = 12$                    $N = 60$                    $N = 600$

time

initial state: block of $N$ particles

Figure 2: Emergence of a limit-shape phenomenon as the system size grows: the particle trajectories are plotted in spacetime, starting from an initial configuration consisting of a single block of $N$ particles on the infinite line. The initial configuration is shown below the first figure on the left. The vertical axis denotes time. Alternating colours are used solely to distinguish neighbouring trajectories. From left to right: $N = 12$, 60, and 600 particles, respectively. We clearly see the emergence, as $N$ increases, of a sharp boundary between 'frozen' regions where the density is 0 or 1, and 'fluctuating' regions where the density is strictly between 0 and 1.

hopping rates Eq. (4) reduces to:

$$w_N^\pm(i) = \frac{1}{2} \prod_{\substack{j=1 \\ j \neq i}}^{N} \left[ 1 \mp \frac{1}{k_j - k_i} \right] \tag{49}$$

. We focus on a single block initial state, with all sites $-\frac{N}{2} + 1, -\frac{N}{2} + 2, \ldots, \frac{N}{2}$ filled with $N \gg 1$ particles. Here we assume that $N$ is even, but this is not essential. It is convenient to introduce the rescaled position and time

$$\xi = \frac{x}{N} \qquad \tau = \frac{t}{N}, \tag{50}$$

so that the initial hydrodynamic density profile is ($N \gg 1$)

$$\rho(\xi, \tau = 0) = \begin{cases} 1, & -\frac{1}{2} \leq \xi \leq \frac{1}{2}, \\ 0, & \text{otherwise.} \end{cases} \tag{51}$$

As we have seen in the previous paragraph, the hydrodynamic profile at later times is given by $\rho(\xi, \tau) = \frac{1}{\pi} \text{Re} \, P(\xi, \tau)$, where $P(\xi, \tau)$ is the solution of (see Eq. (45))

$$P = P_0(\xi + i\tau \sin(P)), \tag{52}$$

where now $P_0(z)$ is given by the $L \to \infty$ limit of formula (47),

$$P_0(z) = i \ln \left( \frac{z + \frac{1}{2}}{z - \frac{1}{2}} \right). \tag{53}$$

Let $Z := e^{iP}$. Then we can rewrite equation (52) as

$$Z = \frac{\tau \left( Z - \frac{1}{Z} \right) + 2\xi - 1}{\tau \left( Z - \frac{1}{Z} \right) + 2\xi + 1}, \tag{54}$$

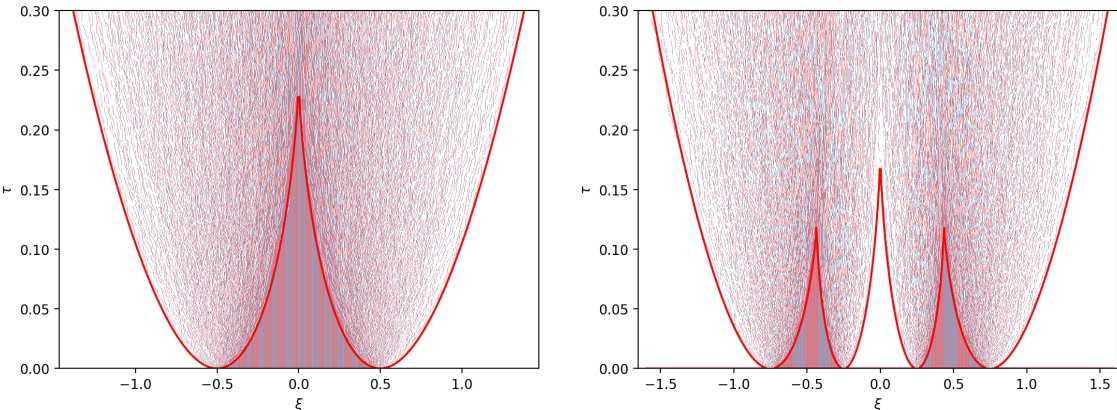

Figure 3: The Arctic curve delineates the frozen regions, where the density is one inside and zero outside. Numerical simulation of particle trajectories from an initial configuration of a single block of $N = 400$ particles centered at the origin (left) and two blocks each containing $N/2 = 200$ particles (right)

which is equivalent to a third-order polynomial equation,

$$(Z-1)(-\tau Z^2 + \tau - 2\xi Z - Z) - 2Z = 0 \tag{55}$$

Since the polynomial in the l.h.s has real coefficients, it has at least one real solution and at most two complex solutions that are complex conjugate. Given that $Z = \exp(i\pi\rho - \pi\tilde{\rho})$, we see that a density $\rho$ that is strictly between 0 and 1 corresponds to having a complex solution with $\mathrm{Im}(Z) > 0$. This requires the discriminant of the above polynomial to satisfy $\Delta < 0$. Accordingly, we should have

$$\begin{aligned} \Delta > 0 \quad &: \quad \text{frozen region,} \quad \text{i.e. } \rho = 0 \text{ or } \rho = 1 \\ \Delta < 0 \quad &: \quad \text{fluctuating region,} \quad \text{i.e. } 0 < \rho < 1. \end{aligned} \tag{56}$$

To find the 'arctic curve' separating the two regions, we must solve the equation $\Delta = 0$. The discriminant of the polynomial (55) is

$$\Delta = -64\tau^3 + 16\tau^2\xi^2 + 48\tau^2 - 80\tau\xi^2 - 12\tau + 16\xi^4 - 8\xi^2 + 1, \tag{57}$$

and by solving $\Delta = 0$ we find the arctic curve plotted in Fig. 3 (left). The agreement with the emergent shape visible from the particle trajectories in the SDEP is compelling.

We note that at large $\tau$ the curve $(\xi, \tau)$ defined by $\Delta = 0$ is asymptotically the parabola $\xi(\tau) \simeq 2\sqrt{\tau}$. Furthermore, one sees that the asymptotic density profile is a semi-circle that expands diffusively,

$$\rho(\xi, \tau) \underset{\xi, \tau \gg 1}{\simeq} \frac{1}{2\pi\tau}\sqrt{4\tau - \xi^2}. \tag{58}$$

To show this, notice first that in the limit of large $\tau$ and $\xi$ Eq. (54) reduces to $Z \approx 1$. This suggests to perform an expansion $Z = 1 + iP - P^2/2 + O(P^3)$. Inserting into Eq. (55) and keeping up to second order, we get:

$$2 + i(3 + 2\xi)P - (5/2 + 2\tau + 3\xi)P^2 = O(P^3). \tag{59}$$

In order for this equation to have solutions, we need $P = O(\tau^{-1/2})$. Set $P = \frac{U}{\sqrt{\tau}}$ and $\xi = 2\gamma\sqrt{\tau}$, with $\gamma \in (-1, 1)$. Keeping only the leading order in $\tau$ of the previous

equation, we get $U^2 - 2i\gamma U - 1 = O(\tau^{-1/2})$. Since $\mathrm{Re}(U) \geq 0$, we select the solution: $U = -i\gamma + \sqrt{1 - \gamma^2}$, which yields:

$$P = \frac{-i\gamma}{\sqrt{\tau}} + \frac{1}{\sqrt{\tau}}\sqrt{1 - \gamma^2} + O(\frac{1}{\tau}), \quad \gamma = \frac{\xi}{2\sqrt{\tau}}. \tag{60}$$

Taking the real part of both sides, we recover Eq. (58).

We note that the expanding semi-circle (58) is also found in the continuous Dyson gas [52]. The reason why we recover the continuous gas behavior here is clear: by letting a block expand along the infinite line ($L \to \infty$), the density ultimately becomes very low, so our lattice gas becomes equivalent to a continuous gas, as discussed in Sec. 3.4.

### 4.2.2 Arctic curve for two blocks.

Similar computations as in the previous paragraph can be done for the initial state with two blocks. Again, for simplicity we focus on the thermodynamic limit $L \to \infty$, and for concreteness we consider the following two initial blocks of particles occupying the sites $-\frac{3N-2}{4}, -\frac{3N-2}{4} + 1, \ldots, -\frac{N+2}{4}$ and $\frac{N+2}{4}, \frac{N+2}{4} + 1, \ldots, \frac{3N-2}{4}$ respectively. Here we are assuming that $N$ is of the form $N = 4p + 2$ for some integer $p$.

We again use the scaling (50) for position and time, then the initial hydrodynamic density is

$$\rho(\xi) = \begin{cases} 1, & \frac{1}{4} \leq |\xi| \leq \frac{3}{4}, \\ 0, & \text{otherwise.} \end{cases} \tag{61}$$

To find the hydrodynamic density at later times, we again need to solve Eq. (52), but this time with

$$P_0(z) = i \ln \left( \frac{z - \frac{1}{4}}{z - \frac{3}{4}} \frac{z + \frac{3}{4}}{z + \frac{1}{4}} \right), \tag{62}$$

which is the $L \to \infty$ limit of Eq. (48).

Like in the single block case, Eq. (52) turns out to be equivalent to a polynomial equation for the variable $Z := e^{iP}$,

$$4\tau^2(Z+1)^2(Z-1)^3 + 4\tau Z (Z^2 - 1)(4\xi(Z-1) + Z + 1) + Z^2(8\xi(2\xi(Z-1) + Z + 1) - 3Z + 3) = 0. \tag{63}$$

Then the arctic curve that separates the 'frozen' region of spacetime —where the density is 0 or 1— from the 'fluctuating' region —where $0 < \rho < 1$— is the set of points $(\xi, t)$ where the discriminant of this polynomial vanishes. The discriminant (not shown here due to its length) can be obtained with the aid of symbolic computation software. The resulting equation is then solved numerically, yielding the curve shown in Fig. 3 (right). Again, we see that the agreement with the stochastic simulation of the SDEP is compelling. Performing a leading-order balance analysis on the discriminant yields an asymptotic arctic curve identical to the single-block case: $\xi = 2\sqrt{t} + O(1)$

## 5 Conclusion

We have shown that the widely held expectation that reversible interacting particle systems generically display deterministic local hydrodynamics under *diffusive* scaling fails for the reversible symmetric Dyson exclusion process (SDEP) - a lattice gas with long-range interactions obtained by conditioning the SSEP on maximal activity and the discrete analogue of the Dyson log-gas. It possesses genuinely non-local hydrodynamics under *Eulerian*

scaling whose low-density limit reproduces the continuum Dyson equation, yet exhibits intriguing limit-shape (arctic-curve) phenomena that spread diffusively at large macroscopic space-time scales and which are absent in the continuous case.

Several open directions now suggest themselves. First, uncovering a genuinely "non-local Kardar-Parisi-Zhang (KPZ)" regime would be particularly exciting. Introducing a weak asymmetry or external noise is a natural route, yet it remains unclear which—if any—perturbations actually generate KPZ-type scaling. A link between free fermions to which the SDEP maps and the KPZ universality class has been pointed out recently [68]. At the PDE level, a Hilbert-kernel variant of the KPZ equation has already been proposed earlier [69]; clarifying its connection to the SDEP is an open problem. Second, applying macroscopic-fluctuation theory, as recently done for the continuum Dyson [51] gas, should reveal how the hard-core constraint alters current large deviations; the long-range MFT developed in [70] offers a natural starting point. Third, coupling the SDEP to particle reservoirs could help us explore basic questions such as how the boundary-induced phase transitions that are well understood for short range interactions [71] would become in the presence of long range interacting. A few promising attempts are already present in the literature [72, 73]. Fourth, a full exploitation of the XX-chain mapping promises refined analytic control. This direction is currently being developed in ongoing work. Finally, conditioning the SSEP on finite (rather than maximal) activity, which corresponds to a Doob transform of the XXZ generator, remains a challenging but exciting avenue.

# Acknowledgments

We thank Kirone Mallick for useful discussions. This work is supported by ANR-PRME Uniopen (project ANR-22-CE30-0004-01), FCT (Portugal) through project UIDB/04459/2020 (10.54499/UIDB/04459/2020) and UIDP/04459/2020 (10.54499/UIDP/04459/2020), and by the FCT Grants 2020.03953.CEECIND/CP1587/CT0013 (10.54499/2020.03953.CEECIND/CP1587/CT0013) and 2022.09232.PTDC (10.54499/2022.09232.PTDC). We acknowledge resources from Mésocentre EXPLOR of the University of Lorraine (project 2024CPMXX3457).

# A  Conditioning a continuous-time Markov process on large deviations

We provide a brief overview of how to condition a continuous-time Markov jump process on an atypical value of a time-additive observable using an exponential tilt and the associated Doob transform. This conditioning involves multiple steps, that we will go through one by one. For detailed reviews see [17, 18, 74].

**Set-up and convention:** Let $(X_t)_{t \geq 0}$ be a continuous-time Markov chain on a finite state space $\Omega$ with generator defined by an intensity matrix $M \in \mathbb{R}^{|\Omega| \times |\Omega|}$. We adopt the *column* convention: for $i \neq j$

$$M_{ij} > 0 \quad \text{is the rate of the jump } j \to i, \qquad M_{ii} = -\sum_{k \neq i} M_{ki}.$$

Thus $\sum_i M_{ij} = 0$ for each $j$, and the forward master equation is $\dot{\mathbf{P}}(t) = M \mathbf{P}(t)$ for the column probability vector $\mathbf{P}(t)$.

**Time-additive observable:** Let $O \subset \{(i, j) \in \Omega \times \Omega : i \neq j\}$ be a specified set of ordered pairs of states (interpreted as jumps $i \to j$). Define the (extensive) counting process

$$J_t = \#\{\, u \in (0, t] : (X_{u^-}, X_u) \in O \,\}.$$

This variable $J_t$ increases by one each time the Markov chain performs the a transition from the set $O$. See [74] for a detailed discussion. The associated *time-averaged counting variable* is

$$j_t := \frac{J_t}{t}.$$

By ergodicity and the law of large numbers $j_t \to \bar{j}$ almost surely as $t \to \infty$, where $\bar{j}$ is the typical value determined by the stationary distribution of $M$ and the rates of jumps in $O$.

**Scaled cumulant generating function:** To probe large fluctuations of $J_t$ (equivalently $j_t$), introduce the scaled cumulant generating function (SCGF)

$$\lambda(s) := \lim_{t \to \infty} \frac{1}{t} \log \mathbb{E}\big[e^{sJ_t}\big]. \tag{64}$$

Provided this limit exists and is differentiable in a neighbourhood of interest, the random variables $j_t = J_t/t$ satisfy a large deviation principle

$$\mathbb{P}(j_t \approx u) \asymp e^{-t\,I(u)} \qquad (t \to \infty),$$

with rate function given by the Legendre–Fenchel transform

$$I(u) = \sup_{s \in \mathbb{R}} \{\, s\,u - \lambda(s) \,\}.$$

**Tilted generator:** Introduce the indicator function $f : \Omega \times \Omega \to \{0, 1\}$ of the set $O$:

$$f(i, j) := \begin{cases} 1, & (i, j) \in O, \\ 0, & \text{otherwise.} \end{cases} \tag{65}$$

We define the *tilted generator* $M^{(s)}$:

$$M_{ij}^{(s)} = M_{ij}\, e^{sf(j,i)}, \qquad i \neq j, \tag{66}$$

Since $f(i, i) = 0$, the diagonal is left unchanged:

$$M_{ii}^{(s)} = M_{ii} = -\sum_{k \neq i} M_{ki}.$$

The matrix $M^{(s)}$ is generally no longer a Markov generator (its column sums are not zero), but it is a real Metzler matrix (nonnegative off-diagonals). For every state $i \in \Omega$ define

$$\psi_i(t; s) \ := \ \mathbb{E}\Big[e^{sJ_t} \,\Big|\, X_t = i\Big], \tag{67}$$

This is the generating function of the current $J_t$ *given that the process ends in state $i$ at time $t$.*

Collecting the components produces a *row* vector,

$$\boldsymbol{\psi}(t; s) \ := \ \big(\psi_i(t; s)\big)_{i \in \Omega}^{\top} \ \in \ \mathbb{R}^{|\Omega|}. \tag{68}$$

Because we have conditioned on the terminal state, this vector evolves according to the *backward* (adjoint) Kolmogorov equation:

$$\frac{\mathrm{d}}{\mathrm{d}t}\,\boldsymbol{\psi}(t;s) \;=\; -\,\widehat{\boldsymbol{\psi}}(t;s)\,M^{(s)}, \qquad \boldsymbol{\psi}(0;s) = \mathbf{1}^{\top}, \tag{69}$$

where $\mathbf{1}$ is an all-ones column vector. The terminal condition reflects the fact that at time $t = 0$ (when no time has elapsed) the observable $J_0$ is zero and the chain must be in its realised final state, so the conditional expectation equals 1.

Solving (69) gives:

$$\boldsymbol{\psi}(t;s) \;=\; \mathbf{1}^{\top}\,\exp\!\big(t\,M^{(s)}\big), \tag{70}$$

Starting from an initial state $P_0$, the unconditional generating function is given by:

$$\mathbb{E}\big[e^{sJ_t}\big] = \mathbf{1}^{\top} e^{\,tM^{(s)}}\mathbf{p}_0,$$

**Spectral representation**    Assume the original chain is irreducible; then, for $s$ in a neighbourhood of 0 (and more generally whenever $M^{(s)}$ remains irreducible), the Perron–Frobenius theorem ensures that $M^{(s)}$ has a unique real, algebraically simple eigenvalue $\Lambda(s)$ with largest real part and strictly positive left and right eigenvectors $\mathbf{l}(s)$ and $\mathbf{r}(s)$:

$$\mathbf{l}(s)^{\top}M^{(s)} = \Lambda(s)\,\mathbf{l}(s)^{\top}, \tag{71}$$

$$M^{(s)}\mathbf{r}(s) = \Lambda(s)\,\mathbf{r}(s), \tag{72}$$

which we normalise by $\mathbf{l}(s)^{\top}\mathbf{r}(s) = 1$. Then

$$\mathbb{E}\big[e^{sJ_t}\big] = \big[\mathbf{1}^{\top}\mathbf{r}(s)\big]\,\big[\mathbf{l}(s)^{\top}\mathbf{p}_0\big]\,e^{t\Lambda(s)}\,[1 + o(1)], \qquad t \to \infty,$$

and hence from (64) we identify

$$\lambda(s) = \Lambda(s).$$

Moreover $\lambda(0) = 0$ and $\mathbf{l}(0) = \mathbf{1}$.

**Doob transform:**    Let

$$\Delta^{(s)} := \mathrm{diag}\big(l_1(s),\ldots,l_{|\Omega|}(s)\big).$$

Define the *Doob-transformed* generator

$$M^{\mathrm{cond}}(s) := \Delta^{(s)}M^{(s)}(\Delta^{(s)})^{-1} - \lambda(s)\,I. \tag{73}$$

Because $\mathbf{l}(s)^{\top}M^{(s)} = \lambda(s)\mathbf{l}(s)^{\top}$, it follows that the columns of $M^{\mathrm{cond}}(s)$ sum to zero and its off-diagonal entries are nonnegative; hence $M^{\mathrm{cond}}(s)$ is a valid CTMC generator (the "driven" or "conditioned" process).

Componentwise for $i \neq j$,

$$M_{ij}^{\mathrm{cond}}(s) = \frac{l_i(s)}{l_j(s)}\,M_{ij}^{(s)} = \frac{l_i(s)}{l_j(s)}\,M_{ij}\,e^{sf(j,i)}, \tag{74}$$

and

$$M_{jj}^{\mathrm{cond}}(s) = M_{jj}^{(s)} - \lambda(s) = M_{jj} - \lambda(s). \tag{75}$$

**Conditioned dynamics** Let $\mathbf{P}(t)$ evolve under the original process: $\dot{\mathbf{P}}(t) = M\,\mathbf{P}(t)$. Define the reweighted measure

$$\mathbf{P}^{\mathrm{cond}}(t) = \frac{\Delta^{(s)}\mathbf{P}(t)}{\mathbf{1}^{\top}\Delta^{(s)}\mathbf{P}(t)}.$$

Differentiating and using (73) shows that

$$\frac{d}{dt}\mathbf{P}^{\mathrm{cond}}(t) = M^{\mathrm{cond}}(s)\,\mathbf{P}^{\mathrm{cond}}(t),$$

so $\mathbf{P}^{\mathrm{cond}}$ evolves according to the Doob-transformed generator. (This equality holds exactly when the normalising scalar is tracked; see, e.g., [17, 18].)

**Invariant measure** Let $\mathbf{r}(s)$ be the positive right Perron eigenvector of $M^{(s)}$:

$$M^{(s)}\mathbf{r}(s) = \lambda(s)\,\mathbf{r}(s).$$

Then

$$M^{\mathrm{cond}}(s)\left(\Delta^{(s)}\mathbf{r}(s)\right) = \left(\Delta^{(s)}M^{(s)}(\Delta^{(s)})^{-1} - \lambda(s)I\right)\Delta^{(s)}\mathbf{r}(s) = \Delta^{(s)}M^{(s)}\mathbf{r}(s) - \lambda(s)\Delta^{(s)}\mathbf{r}(s) = \mathbf{0},$$

so $\Delta^{(s)}\mathbf{r}(s)$ is in the kernel of $M^{\mathrm{cond}}(s)$. Writing it componentwise we obtain the stationary (invariant) distribution

$$\pi_i^{\mathrm{cond}}(s) = \frac{l_i(s)r_i(s)}{\sum_k l_k(s)r_k(s)}.$$

With our normalisation $\mathbf{l}(s)^{\top}\mathbf{r}(s) = 1$, the denominator is 1 and $\pi_i^{\mathrm{cond}}(s) = l_i(s)r_i(s)$.

**Mean current and derivative of the SCGF** Differentiate the right-eigenvector relation $M^{(s)}\mathbf{r}(s) = \lambda(s)\,\mathbf{r}(s)$ with respect to $s$:

$$M'^{(s)}\mathbf{r}(s) + M^{(s)}\mathbf{r}'(s) = \lambda'(s)\mathbf{r}(s) + \lambda(s)\mathbf{r}'(s).$$

Left-multiply by $\mathbf{l}(s)^{\top}$ and use $\mathbf{l}(s)^{\top}M^{(s)} = \lambda(s)\mathbf{l}(s)^{\top}$ and $\mathbf{l}(s)^{\top}\mathbf{r}(s) = 1$ to get

$$\lambda'(s) = \mathbf{l}(s)^{\top}M'^{(s)}\mathbf{r}(s).$$

Because only off-diagonals of $M^{(s)}$ depend on $s$,

$$M_{ij}'^{(s)} = f(j,i)M_{ij}^{(s)}, \qquad i \neq j; \qquad M_{ii}'^{(s)} = 0.$$

Therefore

$$\lambda'(s) = \sum_{i \neq j} f(j,i)\,l_i(s)M_{ij}^{(s)}r_j(s).$$

Use (74) to substitute $M_{ij}^{(s)} = (l_j(s)/l_i(s))M_{ij}^{\mathrm{cond}}(s)$:

$$\lambda'(s) = \sum_{i \neq j} f(j,i)\,M_{ij}^{\mathrm{cond}}(s)\,l_j(s)r_j(s) = \sum_j \pi_j^{\mathrm{cond}}(s)\sum_{i \neq j}M_{ij}^{\mathrm{cond}}(s)\,f(j,i).$$

Recalling that under our column convention $M_{ij}^{\mathrm{cond}}(s)$ is the rate of $j \to i$ in the conditioned process, the inner sum is the *expected instantaneous rate of jumps from $j$ into $O$*. Hence $\lambda'(s)$ is the mean rate of $O$-jumps in the stationary conditioned dynamics. By the ergodic theorem (law of large numbers) for the conditioned CTMC,

$$\lim_{t \to \infty} \frac{J_t}{t} = \lambda'(s) \qquad \text{almost surely under } M^{\mathrm{cond}}(s) \tag{76}$$

Thus the conjugate parameter $s$ continuously controls the typical current in the driven (conditioned) process.

**Relation to micro-canonical conditioning:** Let $j$ be a desired current density. Suppose the LDP rate function $I(u)$ is differentiable at $u = j$. Then the equation

$$\lambda'(s) = j$$

has a unique solution $s = s(j)$, and the long-time path measure of the original process conditioned on $J_t/t \approx j$ (microcanonical) is *logarithmically equivalent* to the path measure of the Doob-transformed process $M^{\text{cond}}(s(j))$ (canonical). That is, bulk observables computed in either ensemble agree in the $t \to \infty$ limit; see [17, 18] and [75] for details.

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
