# Peer review of "Emergent Hydrodynamics in an Exclusion Process with Long-Range Interactions"

_SciPost Physics_

## Round 1 · Referee Report · Anonymous (Referee 1) · 2025-10-5

Strengths

(1) hydrodynamic limit with long-range interactions
(2) Doob transform in the context of stochastic particle systems.
(3) Derivation of novel hydrodynamic equations.
(4) Validation through Monte Carlo simulations.

Weaknesses

None

Report

The authors study the hydrodynamic limit of a one-dimensional stochastic particle system with long-range interactions. This is a very interesting, not so much investigated problem. Their
detailed analytic results are based on a Doob transform yielding an integrable quantum system. The novel results provide important insights on the special feature of long-range forces. While Doob
transform as such is a known method, the particular application to stochastic many-particle systems is novel. With great care, the predictions on macroscopic behavior are validated by numerical simulations.

The article is very well written and readable for a larger community.

In view of the novelty and density of results I recommend publication in SciPost Physics.

Requested changes

I have only two comments:

(1) Eq. (32) is a coupled system, so why independent.

(2) Since the interaction is long-ranged, the local jump rates might be defined only for a restricted class of initial conditions. Maybe I overlooked, but there does not seem to be a discussion.

Recommendation

Publish (surpasses expectations and criteria for this Journal; among top 10%)

---

## Editorial Decision

in_refereeing